# Drivers of intermodel uncertainty in land carbon sink projections

Ryan S. Padrón[1], Lukas Gudmundsson[1], Laibao Liu[1], Vincent Humphrey[1,2], Sonia I. Seneviratne[1]

[1]Institute for Atmospheric and Climate Science, Department of Environmental Systems Science, ETH Zurich, Zurich, 8092, Switzerland

[2]Department of Geography, University of Zurich, Zurich, Switzerland

*Correspondence to*: Ryan S. Padrón (ryan.padron@env.ethz.ch)

**Abstract.** Over the last decades, land ecosystems removed from the atmosphere approximately one third of anthropogenic carbon emissions, highlighting the importance of the evolution of the land carbon sink for projected climate change. Nevertheless, the latest cumulative land carbon sink projections from eleven Earth system models participating in the

Coupled Model Intercomparison Project Phase 6 (CMIP6) show an intermodel spread of 151 PgC (i.e., ~15 years of current anthropogenic emissions) for a policy-relevant scenario with mean global warming by the end of the century below 2°C relative to preindustrial conditions. We hypothesize that this intermodel uncertainty originates from model differences in the sensitivity of photosynthesis to atmospheric $CO_2$ concentration (i), the sensitivity of net biome production (NBP) to air temperature (ii) and soil moisture (iii), as well as model differences in average conditions of air temperature (iv) and soil

moisture (v). Using multiple linear regression and a resampling technique, we quantify the individual contributions of these five terms for explaining the cumulative NBP anomaly of each model relative to the multi-model mean. Results indicate a primary role of the response of NBP to interannual temperature and soil moisture variability, followed by the sensitivity of photosynthesis to $CO_2$, and lastly by the average climate conditions, which also show sizeable contributions. We find that the sensitivities of NBP to temperature and soil moisture, particularly in the tropics, dominantly explain the deviations from the

ensemble mean of the two models with the lowest carbon sink (ACCESS-ESM1-5 and UKESM1-0-LL) and of the two models with the highest sink (CESM2 and NorESM2-LM). Overall, this study provides insights on why each Earth system model projects either a low or high land carbon sink globally and across regions relative to the ensemble mean, which can focalize efforts to identify the representation of processes leading to intermodel uncertainty.

## 1 Introduction

During recent decades, ecosystems on land have taken up approximately one third of anthropogenic carbon emissions to the atmosphere, and are predominantly responsible for the year-to-year variations in the atmospheric carbon growth rate (Friedlingstein et al., 2020). While carbon emissions have been on the rise, land ecosystems have taken up more and more carbon, which has resulted in this approximately constant 30% sink (Canadell et al., 2021). However, it remains unclear to what level this capacity of land to remove carbon from the atmosphere can continue in the coming decades, with evidence

pointing towards a less effective sink under increasing cumulative emissions (Gatti et al., 2021; Hubau et al., 2020; Koch et

al., 2021; Raupach et al., 2014; IPCC, 2021). The future evolution of the land carbon sink is crucial to project how much global warming and consequent climate change Earth will experience given a certain level of greenhouse gas emissions. It is thus linked to policy questions such as how much more carbon can we emit to limit global warming below the 2°C or 1.5°C thresholds decided in the 2015 Paris agreement.


The net carbon flux from the atmosphere to the land is denoted as net biome production (NBP), and it is given by the carbon uptake of vegetation through photosynthesis (gross primary production (GPP)), minus the losses of carbon to the atmosphere through autotrophic (RA) and heterotrophic respiration (RH), as well as from ecosystem disturbances (DIS) such as fires. These fluxes are influenced by both atmospheric $CO_2$ concentration and climate conditions. Rising atmospheric $CO_2$
primarily favors higher GPP (fertilization effect), while it can indirectly enhance RA due to greater plant biomass and enhance RH due to higher microbial decomposition of fresh carbon supplied by increased litterfall and root-derived labile soil carbon, as well as higher priming of old soil organic matter fueled by this increased supply of labile soil carbon (Gao et al., 2020). Temperature conditions can influence all fluxes, with warming notably leading to an increase in respiration (Varney et al., 2020). In addition, the relevance of water-carbon interactions for the land carbon cycle has gained recognition
in recent years (Canadell et al., 2021; Gentine et al., 2019; Green et al., 2019; Huang et al., 2020; Humphrey et al., 2018; Liu et al., 2020; Novick et al., 2016; Peters et al., 2018; Stocker et al., 2018). Observations indicate a high sensitivity of annual NBP to anomalies in terrestrial water storage at the global scale, which is underestimated by land surface models uncoupled from the atmosphere (Humphrey et al., 2018; Peters et al., 2018). Other factors such as incoming radiation and vapor pressure deficit (air dryness) clearly influence GPP and the land carbon sink (Fu et al., 2022; Grossiord et al., 2020;
Humphrey et al., 2021; Novick et al., 2016), nevertheless they are strongly correlated with temperature and soil moisture on monthly or longer time scales. Overall, $CO_2$ fertilization increases global land NBP, while anthropogenic warming and associated climate change tend to reduce it (Arora et al., 2020; Canadell et al., 2021). Importantly, the response of NBP to interannual climate variability appears to be underestimated by multiple Earth system models, potentially implying a smaller than expected capacity of land to remove carbon from the atmosphere during this century (Fu et al., 2022; Humphrey et al.,
2018; Novick et al., 2016; Winkler et al., 2021).

Projections of the land carbon sink from Earth system models have shown a very substantial intermodel uncertainty since early Coupled Model Intercomparison Projects (CMIPs) (Bonan and Doney, 2018; Friedlingstein et al., 2006) and this continues to be the case for the latest models participating in CMIP6 (Arora et al., 2020; Canadell et al., 2021). While
uncertainty in the projections is dominated by these intermodel differences, internal climate variability can also play an important role (Tokarska et al., 2020). As models continue to evolve, land carbon cycle processes can be represented differently across them, and some models may represent processes that others do not, contributing to the uncertainty. Phenology (Peano et al., 2021), water transport through vegetation and water stress (Lawrence et al., 2019), fire (Hantson et al., 2020), woody-plant mortality (De Kauwe et al., 2022; McDowell et al., 2022), and nutrient limitation (Davies-Barnard et

al., 2020) are processes at the modelling frontier. The representation of the nitrogen cycle in more models was an important step forward going from CMIP5 to CMIP6 (Davies-Barnard et al., 2020). It is often the case that land carbon uptake is reduced when including nitrogen availability as a constraint (Arora et al., 2020; Canadell et al., 2021). In addition, several studies have proposed observational constraints for projections of different land carbon sink components and sensitivities (Cox et al., 2013; Liu et al., 2019; Mystakidis et al., 2016, 2017; Schlund et al., 2020; Varney et al., 2020). Despite this progress, land biogeochemical feedbacks with climate change remain a major source of uncertainty for future carbon budgets.

The typical framework to study the evolution of the land carbon cycle and associated intermodel uncertainty is based on the carbon-concentration and carbon-climate feedback parameters (Arora et al., 2013, 2020; Friedlingstein et al., 2006). These feedback parameters are estimated from idealized simulations that increase atmospheric $CO_2$ concentration 1% per year until 2-times (~560 ppm) or 4-times (~1120 ppm) its pre-industrial value and that distinguish between its radiative and biogeochemical effects. Using these simulations, Arora et al. (2020) provide insights on why the global land carbon feedback parameters differ among models by comparing their changes in vegetation and soil carbon pools, their strength of the $CO_2$ fertilization effect on GPP, residence time, allocation, and changes in carbon use efficiency globally. To complement these insights and further our understanding of the drivers of intermodel uncertainty in land carbon sink projections we directly focus on local responses to $CO_2$ concentration, temperature, and soil moisture under the SSP126 low emission scenario (peak concentration of 471 ppm around 2075 and 446 ppm by 2100 (Meinshausen et al., 2020)).

In this study we aim to advance our understanding of the future evolution of the land carbon cycle, primarily for a policy-relevant scenario where global warming is limited to below 2°C. Here we use an ensemble of Earth system models participating in CMIP6, as detailed in section 2. A global and spatially explicit overview of the differences in NBP across models is given in section 3. Section 4 describes across the model ensemble the following five proposed drivers of projected cumulative NBP: (i) sensitivity of gross primary production (GPP) to $CO_2$ concentration, (ii) sensitivity of NBP to air temperature, (iii) sensitivity of NBP to soil moisture, (iv) long-term average temperature, and (v) long-term average soil moisture. In section 5 we quantify the contribution of differences in each of these drivers for explaining the intermodel differences in cumulative NBP. Concluding remarks are provided in section 6.

## 2 Model simulations and characteristics

For the main analysis we employ all 11 models participating in CMIP6 that provide NBP, near surface air temperature and layered soil moisture data for the SSP126 scenario (Table S1), which are publicly available at https://esgf-node.llnl.gov/search/cmip6/. Data from all models are regridded to a common a 2.5°×2.5° longitude-latitude grid using second order conservative remapping, and a land mask is applied to increase model comparability. The SSP126 scenario is a

shared socio-economic pathway based on the world following an ecological transition (Riahi et al., 2017), for which global warming by the end of the century is projected to be less than 2°C relative to preindustrial conditions. In addition, to estimate the sensitivity of GPP to $CO_2$ concentration, we employ the 1pctCO2-bgc simulations in which atmospheric $CO_2$ increases 1% per year and only its direct physiological effects on vegetation are considered, while neglecting the radiative effects (Jones et al., 2016). Information from other scenarios is used in some cases to complement the analysis.

An overview of the carbon cycle representation in the models analyzed in this study is provided in Table S1. Additional information about the models, except CMCC-ESM2 and EC-Earth3-Veg, is given by Arora et al. (2020). From all 11 models analyzed here, ACCESS-ESM1-5 is the only model that includes a phosphorous cycle in addition to nitrogen, whereas CanESM5, GFDL-ESM4 and IPSL-CM6A-LR do not include a nitrogen cycle, and CNRM-ESM2-1 only has an implicit representation of the nitrogen cycle. Fire emissions are not represented in 4 of the 11 models, namely ACCESS-ESM1-5, CanESM5, IPSL-CM6A-LR and UKESM1-0-LL. Dynamic vegetation cover is only modeled by EC-Earth3-Veg, GFDL-ESM4, MPI-ESM1-2-LR and UKESM1-0-LL. In addition, soil moisture storage capacity and discretization of soil layers can be very different across models (Table S2). To increase comparability, here we compute soil moisture down to a depth of 30 cm for every model by summing the moisture of all corresponding layers. For cases when the 30-cm depth threshold is within the boundaries of a model layer, we assume that the moisture in the layer is distributed uniformly with depth and account proportionally for the moisture until the 30 cm threshold.

## 3 Intermodel differences in NBP

The cumulative global land carbon sink projections from eleven Earth system models until the year 2100 show large differences, with an ensemble range between 56.3 PgC and 206.6 PgC, a multi-model mean estimate of 144.7 PgC, and an intermodel standard deviation of 47 PgC (Fig. 1). The intermodel spread of 150.3 PgC corresponds to approximately 40% of the remaining carbon budget to limit global warming below 2°C (with a 50% likelihood) according to Table SPM.2 of IPCC (2021). Even though these differences are rather large it is important to note that they are smaller than those corresponding to higher emission scenarios such as SSP585, where the range is approximately 100–700 PgC. Thus, uncertainty increases as we move further away from the current state of the system to higher concentrations of greenhouse gases in the atmosphere. Differences across models are considerably larger than differences across realizations of individual models, which partly represent the internal variability of the system (Fig. S1).

Further insights on projected intermodel NBP differences are obtained by decomposing them into differences in GPP, autotrophic and heterotrophic respiration, and disturbances (Fig. S2). There is no clear correspondence between models with higher global land GPP having also higher NBP, given that the loss terms are also generally higher. Moreover, the three models with highest NBP project relatively low GPP, as well as low carbon use efficiency (i.e., CUE = (GPP – RA)/GPP),

but heterotrophic respiration and disturbance losses are also low. The CUE ranges from 0.55 for CanESM5 to 0.41 for EC-Earth3-Veg and ACCESS-ESM1-5. Models that exhibit a relatively high CUE tend to also exhibit high heterotrophic respiration, and thus moderate NBP. Lastly, we note that EC-Earth3-Veg and GFDL-ESM4 show the highest disturbance fluxes mainly from fire emissions. Although DIS is rather small compared to GPP, RA and RH, it is still large enough to substantially influence intermodel differences in NBP.

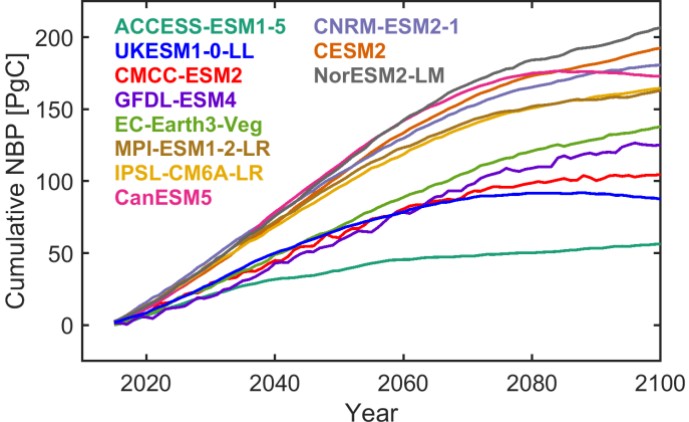

**Figure 1: Intermodel differences in projected global land NBP for scenario SSP126.** Temporal evolution of projected global land cumulative NBP from individual models. The average value is shown for models with multiple realizations available. If a model does not have information over Greenland, NBP is set to zero. Cumulative global land NBP is estimated by multiplying the area-weighted average NBP times the land surface area excluding Antarctica (135.22E6 km$^2$).

The maps of cumulative NBP illustrate in more detail the characteristics of the projections from individual models (Fig. 2). Note that there can be models with similar global land NBP as indicated in Fig. 1, even though the underlying spatial distribution is markedly different. Take MPI-ESM1-2-LR and IPSL-CM6A-LR as an example, whereas the tropics – particularly central Africa – are the main sink in the MPI-ESM1-2-LR model, the northern mid and high latitudes contribute most of the sink in IPSL-CM6A-LR. The UKESM1-0-LL and CMCC-ESM2 models also show a very contrasting spatial pattern of cumulative NBP, despite having a relatively similar global land sink magnitude. The local intermodel standard deviation of projected NBP (bottom right panel of Fig. 2) points towards the tropics and boreal forests as the regions with higher discrepancies across models. In addition, models also show marked differences in the magnitude of the detrended interannual variability of NBP (Fig. S3). EC-Earth3-Veg and GFDL-ESM4 have high interannual variability, as well as CMCC-ESM2 particularly over the boreal forests, whereas CESM2 and NorESM2-LM show the least variability.

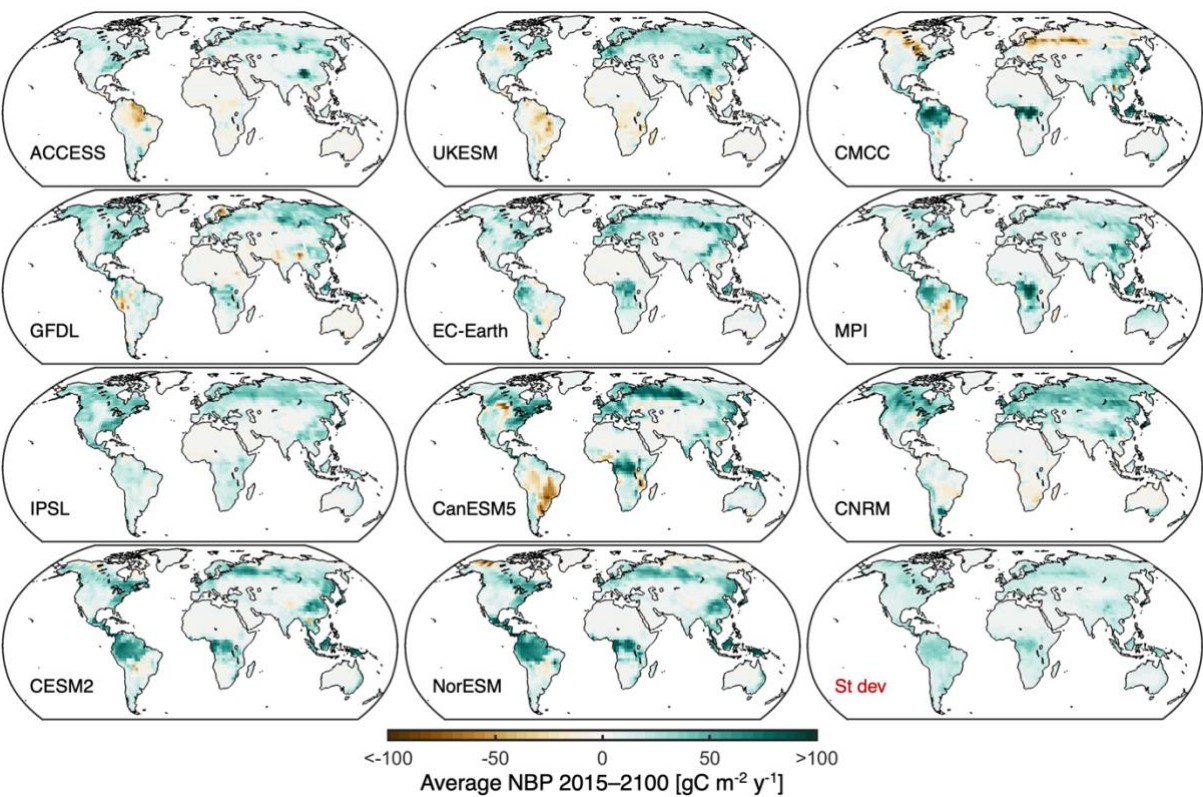

150 **Figure 2: Intermodel differences in regional projected NBP.** Maps of average projected NBP during 2015 to 2100 from individual models. The intermodel standard deviation is shown at the bottom right.

## 4 Drivers of intermodel differences in NBP projections

Several factors can influence long-term NBP, with atmospheric $CO_2$ concentration, air temperature (T) and soil moisture (SM) playing a potentially important role. Here we focus on the sensitivities of NBP to changes in these variables, as well as
155 on the background conditions of T and SM across the model ensemble. Background conditions of atmospheric $CO_2$ are prescribed and thus identical for all models. Other aspects such as land cover, incoming radiation, air humidity, carbon allocation, nutrient constraints, fire emissions, and interactions between $CO_2$ with T and SM can also be relevant for NBP, however they are addressed only indirectly in this study.

### 4.1 Sensitivity of GPP to atmospheric $CO_2$ concentration

160 The physiological effect of atmospheric $CO_2$ concentration on NBP is dominated by its fertilization effect on GPP, while indirect $CO_2$ effects on RA, RH and DIS can also be substantial. Here we use the 1pctCO2-bgc simulations to estimate the sensitivity of NBP to $CO_2$ concentration at every grid cell. These simulations limit confounding effects from changes in temperature and soil moisture as they only account for the biogeochemical effects of rising $CO_2$. However, when computing

the change in NBP in these simulations, it is important to note that model differences can also arise from differences in RA, RH and DIS that are highly dependent on how these fluxes are influenced by temperature and soil moisture in each model. Therefore, we decide to use the sensitivity of GPP (instead of NBP) to $CO_2$ as a driver of intermodel uncertainty in land carbon sink projections to better disentangle the influence of $CO_2$ from that of temperature and soil moisture, even though the indirect effects of $CO_2$ on RA, RH and DIS are ignored in this case. Alternatively, we also replicate the analysis when using the sensitivity of NBP to $CO_2$ instead and present results in the supplement.

The sensitivity of GPP to $CO_2$ ($sCO_2$) is computed as the slope of the linear regression based on the 30 annual values available in the simulations in which $CO_2$ concentration ranges from approximately 375 ppm to 500 ppm. Even though some of these concentrations are outside the range of 400–471 ppm spanned in the SSP126 scenario, the responses of GPP and NBP to $CO_2$ concentration are still mostly linear for all models (Fig. S4). If we were to limit the sample to be within the SSP126 $CO_2$ concentration range, only 17 annual values would be available, which would reduce the confidence in the estimated regression slope. Counting with a larger sample size reduces the potential confounding effect of local temperature and soil moisture anomalies from individual years. To further avoid this confounding effect from individual years, we compute the regression slope one hundred times after resampling with replacement from the 30-year time series. Finally, the median value of these regression slopes is used as the representative sensitivity from each model. In the case of models with multiple 1pctCO2-bgc simulations the average value from all simulations is used. Given that no data is available from NorESM2-LM for the 1pctCO2-bgc simulation, we here assume that its $sCO_2$ is the same as for CESM2 because both share the same land surface model (CLM5).

Increasing atmospheric $CO_2$ concentration favors an overall increase in GPP due to enhanced photosynthesis, although with substantial intermodel differences in $sCO_2$ particularly in the tropics (Fig. S5). EC-Earth3-Veg, CMCC-ESM2, and ACCESS-ESM1-5 generally exhibit the lowest fertilization effect from atmospheric $CO_2$, whereas GFDL-ESM4 and MPI-ESM1-2-LR exhibit the overall highest despite relatively low values at high latitudes. It is surprising that $sCO_2$ is negative at several locations, particularly for EC-Earth3-Veg and ACCESS-ESM1-5. This is likely due to natural climate variability in the simulations rather than an actual negative effect of increasing $CO_2$ on GPP. Other noteworthy features are the relatively low tropical and high extratropical $sCO_2$ of IPSL-CM6A-LR and CNRM-ESM2-1, as well as the relatively high tropical and low extratropical $sCO_2$ of CanESM5.

## 4.2 Sensitivity of NBP to temperature and soil moisture

Locally, annual warm or cold and wet or dry anomalies can influence annual NBP. The interannual sensitivity of NBP is therefore potentially indicative of the consequences of long-term changes in T and SM on the land carbon sink. In addition, an asymmetry in the response of NBP to a warm (dry) and cold (wet) anomaly of equal magnitude would influence cumulative NBP even if there were no long-term changes in T and SM. In this case too, a different sensitivity would

contribute to a difference in cumulative NBP. Thus, model differences in the sensitivity of NBP to interannual variations in T and SM can potentially explain differences in cumulative NBP.

200 We estimate the sensitivity of NBP to temperature (sT) and soil moisture (sSM) from the detrended time series of annual mean NBP and detrended annual mean warm-season T and SM values from 2015 to 2100 given by the SSP126 simulations. Detrending the time series reduces the confounding effect of rising $CO_2$ concentrations in these simulations, although potential alleviating effects of higher $CO_2$ for NBP when facing T and SM anomalies are implicit within sT and sSM. The removed trends are computed using a lowess fit with a 30-year window. In tropical latitudes (i.e., below 22.5° based on the 205 model grid), we consider all months of the year, whereas in higher latitudes, we focus on the warmer months when vegetation is more active: March–October in the Northern Hemisphere and September–April in the Southern Hemisphere. We define sT and sSM as the covariance of NBP with T and SM, as opposed to the regression slope, to also account for intermodel differences in the interannual variability of T and SM (this is not necessary for $sCO_2$ given that the interannual variability of $CO_2$ is prescribed to be the same for all models). In addition to the covariance, we also compute the Pearson 210 correlation to better describe the coupling of NBP with T and SM in the models. In the case of models with multiple simulations the average covariance and average correlation are used.

Anomalies in NBP and T are generally negatively correlated (years with higher T lead to lower NBP) throughout the world for most models, except at high latitudes, whereas the opposite is the case for the correlation with SM (years with higher SM 215 lead to higher NBP) (Fig. 3). Nevertheless, there are several regions where the correlations are weaker and/or model disagreement is higher as indicated by values closer to zero of the multi-model mean over the standard deviation, e.g., southeast Asia, China, central Europe, central Africa (particularly for sSM), and throughout the boreal forests. Maps of the correlations and covariances from individual models are provided in the Supplement (Figs. S6–S9). Some noteworthy features are the highly negative correlation of NBP with T and highly positive correlation with SM over tropical South 220 America for the ACCESS-ESM1-5, UKESM1-0-LL, GFDL-ESM4, EC-Earth3-Veg, and CanESM5 models; the highly positive covariance of NBP with SM over the boreal forests for CMCC-ESM2 and EC-Earth3-Veg; and the negative covariance of NBP with SM in southeast Asia, China, east North America, and southeast Brazil plus Uruguay for UKESM1-0-LL and MPI-ESM1-2-LR.

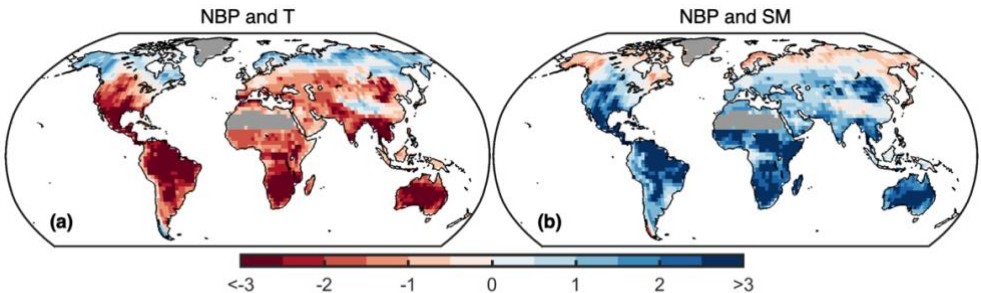

NBP and T      NBP and SM

(a)      (b)

<-3   -2   -1   0   1   2   >3

Ratio of multi-model mean correlation over standard deviation

**Figure 3: Model agreement in the sensitivity of NBP to temperature and soil moisture.** Shown is the ratio of the multi-model mean over the multi-model standard deviation for (a) the correlation between detrended annual NBP and detrended annual warm-season temperature, and for (b) the correlation between detrended annual NBP and detrended annual warm-season soil moisture. Lighter colors indicate weaker correlations and/or higher model disagreement. Regions with 2015–2100 average GPP below 100 gC m$^2$ y$^{-1}$ are masked.

Although positive T anomalies often coincide with negative SM anomalies, we find that for most models and across most regions SM explains the detrended interannual variability of NBP better than T, as indicated by the squared correlations (Fig. 4a). Anomalies in SM explain 50% of the variability in NBP over a quarter of the land area (excluding regions with average GPP below 100 gC m$^2$ y$^{-1}$) on average across all models. Note that in general the interannual correlation of NBP and SM, as well as of NBP and T, is higher precisely where the interannual variability of NBP is higher (Figs. S3, S6 and S7). In addition, to explain the detrended interannual variability of NBP using both T and SM as predictors, we fit a stepwise linear regression. In this case the explained NBP variability increases, reaching an ensemble mean of 57% over a quarter of the land area. Furthermore, we find that over 78% of the land area there is a model majority for which SM is added as the first predictor (Fig. 4b). Notable exceptions are regions with large interannual NBP variability such as parts of the Amazon, central Africa, and southeast Asia, where T is added first for most models. However, note that there is no strong model agreement throughout many regions. Results are consistent when using SM down to 1 m depth instead of 30 cm (Fig. S10). Overall, these findings highlight the importance of explicitly considering the sensitivity to SM, in addition to the sensitivity to T, as a driver of intermodel differences in NBP.

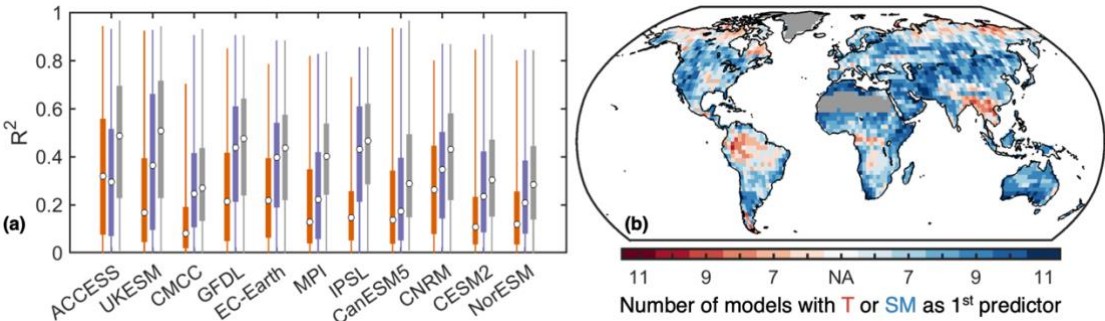

Number of models with T or SM as 1$^{st}$ predictor

11   9   7   NA   7   9   11

**Figure 4: Temperature and soil moisture as drivers of the detrended interannual variability of NBP.** (a) Area-weighted distribution from all grid cells of the coefficient of determination (R$^2$) between NBP with T (orange), with SM (purple), and with both T and SM from a stepwise linear regression (gray). The circle indicates the median, the boxes span the interquartile range, and the whiskers the full range. (b) Model agreement at each location where either T or SM is added first to the stepwise linear regression. Regions with 2015–2100 average GPP below 100 gC m$^2$ y$^{-1}$ are omitted in (a) and masked in (b).

## 4.3 Average warm-season temperature and soil moisture conditions

Projected long-term average T and SM also influence the cumulative land carbon sink. NBP is reduced if conditions are generally too hot, cold, dry, or wet relative to an optimum state. Given that our focus is on cumulative NBP from 2015 to 2100 under the SSP126 scenario, we compute the average warm-season T and SM over the same 86-year period and scenario. We follow the same definition for the warm season as described in section 4.2.

Local intermodel differences in projected average T and top 30 cm SM show a global mean standard deviation of 1.4°C and 14.7 kg m$^{-2}$ over land, with higher values at tropical forests, the United States and Tibet for T, and at very high latitudes for SM (Figs. S11 and S12). GFDL-ESM4, MPI-ESM1-2-LR, IPSL-CM6A-LR and EC-Earth3-Veg generally project lower temperatures, whereas higher temperatures are projected by ACCESS-ESM1-5 and UKESM1-0-LL in the tropics, by NorESM2-LM, CMCC-ESM2 and CESM2 at mid latitudes, and by CanESM5, CMCC-ESM2, IPSL-CM6A-LR and CNRM-ESM2-1 at high latitudes. NorESM2-LM and CESM2 are clearly the models with highest SM (both have CLM5 as their land surface model). On the other hand, MPI-ESM1-2-LR, IPSL-CM6A-LR, CanESM5 and UKESM1-0-LL are generally the driest models, as well as ACCESS-ESM1-5 in the tropics.

## 5 Explaining intermodel differences in cumulative NBP

To assess the effect and relevance of each of the five proposed drivers of intermodel differences in land carbon sink projections, we compute at each grid cell their correlation with cumulative NBP (Fig. 5). At each grid cell, for example, we correlate the $sCO_2$ values obtained for each of the 11 models with the corresponding 11 values of cumulative NBP from each model. The dominant positive correlation for $sCO_2$, particularly in the extratropics, indicates that models that have a higher $CO_2$ fertilization effect on GPP tend to project higher NBP in these regions. Throughout the tropics, the lack of correlation indicates that intermodel differences in NBP cannot be explained by differences in the strength of the $CO_2$ fertilization effect, potentially because of other NBP drivers. In addition, we find clear positive correlations of cumulative NBP with sT and negative correlations with sSM over multiple regions important for the land carbon sink, such as the Amazon, central Africa, India, China, eastern Australia, Europe, and the boreal forests. This suggests that models that have higher (lower) NBP during warmer (colder) than average years (i.e., higher sT) tend to project higher cumulative NBP in these regions, as do models that have higher (lower) NBP during drier (wetter) than average years (i.e., lower sSM). In other words, models for which annual NBP drops less during hotter and drier years yield higher cumulative NBP (recall Fig. 3). On the other hand, there are also other typically wet regions such as Indonesia and southeast South America where models that have higher (lower) NBP during wetter (drier) than average years (i.e., higher sSM) tend to project higher cumulative NBP. Additionally, we find that higher long-term average warm season T in some models over central Africa, eastern Brazil, the Amazon, as well as central and western United States is associated with lower cumulative NBP. Higher NBP is favored over midwestern

North America, the Amazon, European boreal forests, and eastern Australia for models with higher-than-average long-term
mean warm season SM.

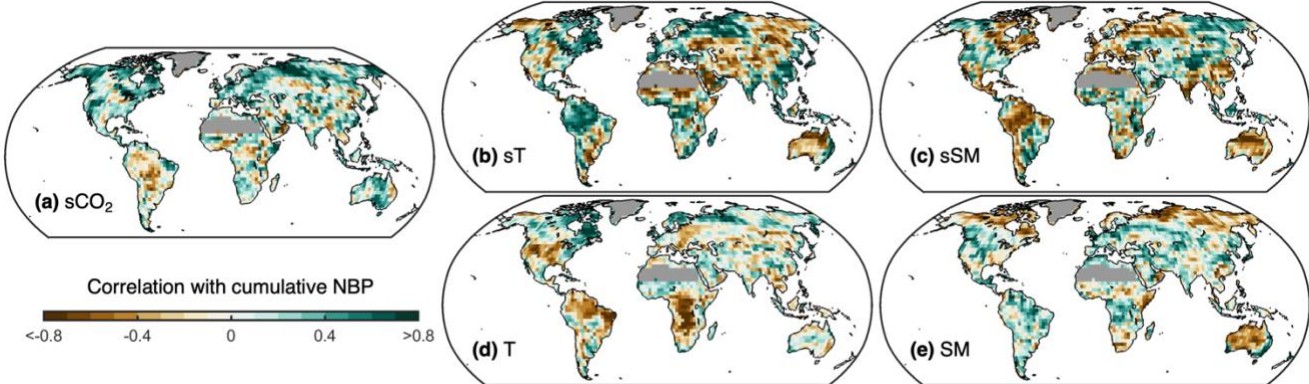

**Figure 5: Influence of model characteristics on projected cumulative NBP.** Pearson correlation at each grid cell between cumulative NBP projected by each Earth system model and their (a) sensitivity to $CO_2$ concentration (sCO2), (b) sensitivity to temperature (sT), (c) sensitivity to soil moisture (sSM), (d) long-term warm season average temperature (T), and (e) long-term warm season average soil moisture (SM). For (b) and (c), sT and sSM correspond either to the covariance or to the correlation of detrended annual anomalies of NBP with T and SM, depending on which has the strongest absolute correlation with projected cumulative NBP. Regions with 2015–2100 average GPP below 100 gC m$^{-2}$ y$^{-1}$ are masked.

To quantify the joint contributions of differences in sCO2, sT, sSM, T and SM for explaining the intermodel differences in projected cumulative NBP, we fit a multiple linear regression at every grid cell. As noted in section 4.2, for sT and sSM we use both the covariance and the correlation of detrended annual anomalies of NBP (NBP$_{anom}$) with anomalies of warm season temperature (T$_{anom}$) and soil moisture (SM$_{anom}$). Therefore, cumulative NBP of each model $m$ is estimated according to Eq. (1):

$$NBP_m = b_0 + b_1 \times sCO_{2\,m} + b_2 \times cov(NBP_{anom}, T_{anom})_m + b_3 \times corr(NBP_{anom}, T_{anom})_m +$$
$$b_4 \times cov(NBP_{anom}, SM_{anom})_m + b_5 \times corr(NBP_{anom}, SM_{anom})_m + b_6 \times T_m + b_7 \times SM_m , \qquad (1)$$

where $b_i$ are the regression coefficients. Given that the number of predictor variables (7) is relatively high, and the sample size is relatively small (11 models), we are likely to obtain a good fit for the regression even if random variables are used instead of the proposed drivers. However, a regional spatial coherence on the signs of $b_i$ would only arise if there is an actual relation between NBP and the proposed drivers as seen in Fig. 5.

Instead of fitting the regression given by Eq. 1 only once at every grid cell using cumulative NBP from 2015 to 2100, we create 200 different bootstrap time series of 86 years by resampling with repetition from all projected years. Then for each of these time series we compute the cumulative NBP of individual models, as well as sT, sSM, T and SM (not sCO2 as it is estimated from the 1pctCO2-bgc simulations), and finally fit the regression. This bootstrap approach introduces some uncertainty in our estimates of both the proposed drivers (Eq. 1, right-hand side) and the cumulative NBP (Eq. 1, left-hand

side), which is further propagated into an uncertainty in terms of the regression coefficients $b_i$. In this way we can provide a likely range for the contribution of each driver to explain a given model's anomalous behaviour relative to the ensemble mean cumulative NBP.

The regression estimates capture well the local intermodel variability of cumulative NBP (Fig. S13) and the regression
coefficients $b_i$ show regionally coherent signs consistently across the bootstrap realizations (Fig. S14), providing confidence in the robustness of the results. The net aggregated outcome of the local multiple linear regressions (MLR) (Eq. 1) shows good agreement with average projected NBP from individual Earth system models (ESM) at regional and global scales (Fig. 6). The average coefficient of determination ($R^2$) across bootstraps between the ESM and MLR aggregated estimates is greater than 0.94 for both the global and regional estimates. Thus, most of the intermodel spread in cumulative NBP can be
explained from the intermodel spread in the proposed drivers. Nevertheless, there are some discrepancies between the MLR estimate and the ESM projection of each model. Most noticeable are the MLR underestimation of the land carbon sink modelled by CanESM5, NorESM2-LM, CESM2 and CNRM-ESM2-1, as well as the overestimation for UKESM1-0-LL, GFDL-ESM4, and CMCC-ESM2. The underestimation for CanESM5 is consistent across all regions, for NorESM2-LM and CESM2 it arises mainly in the tropics, and for CNRM-ESM2-1 it occurs at mid and high latitudes. In the case of UKESM1-
0-LL the overestimation from the regression predominantly occurs in the tropics, and to a lesser degree at high latitudes. For GFDL-ESM4 the overestimation is highest at mid latitudes, and it also occurs in the tropics. For CMCC-ESM2 there is a large overestimation at high latitudes that is partially compensated by an underestimation at the tropics. These differences are likely due to nonlinear responses of NBP that are not captured by the multiple linear regression and/or due to missing relevant drivers (e.g., indirect effects of $CO_2$ on RA, RH and DIS, differences in land cover, etc.).

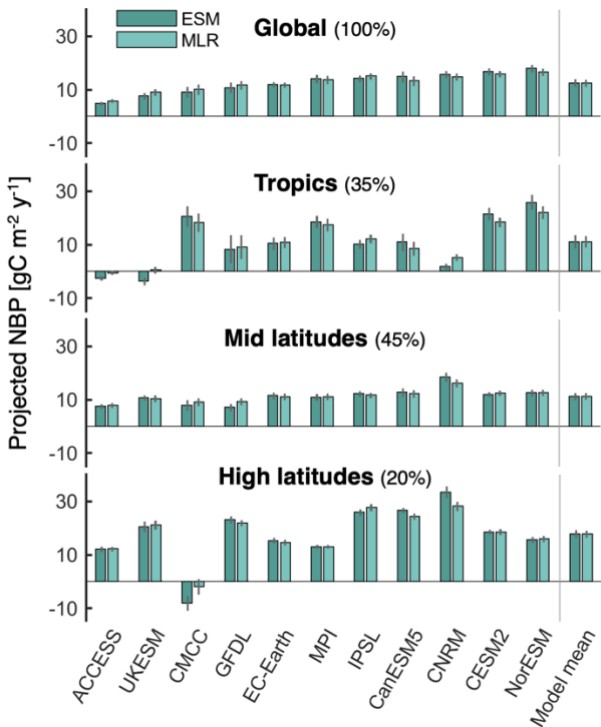


**Figure 6: Comparison of average projected NBP from Earth system models (ESM) with the multiple linear regression (MLR) estimate.** Colored bars indicate the mean from the 200 bootstrap samples, while uncertainty bars span from the 5th to the 95th percentile. The spatial average is shown for global land, the tropics (22.5°S – 22.5°N), mid latitudes (22.5°N – 47.5°N over North America, 22.5°N – 55°N over Europe and Asia, and > 22.5°S), and high latitudes (> 47.5°N over North America and > 55°N over Europe and Asia). The land
percentage comprised in each region is noted next to the title: tropics represent 35.1% of the considered global land area, mid latitudes 44.8%, and high latitudes 20.1%.

Given that the regression simulates well the projected NBP by the Earth system models, the individual terms of Eq. 1 (i.e., $b_i * driver_i$) can be used to quantify the contribution of each proposed driver to explain intermodel differences. The contributions of sT and sSM are obtained by lumping both the correlation and covariance terms. Figure 7 shows how much
each driver contributes to explain each model's anomaly in projected NBP relative to the ensemble mean (see also Fig. S15). For example, 27% of the lower-than-average land carbon sink projected by ACCESS-ESM1-5 is due to its low sCO$_2$ mainly outside the tropics (see Fig. S5), 65% is due to its low sT mainly at the tropics (i.e., lower (higher) NBP during hotter (colder) than average years compared to other models (see Figs. S6 and S8)), 19% is due mainly to its high long-term average tropical T (see Fig. S11), and 7% is due to its SM, whereas the contribution of sSM compensates the excess 18%
from the other drivers. UKESM1-0-LL also shows an important contribution towards a relative low land carbon sink from sT and T mainly at the tropics. At the other end of the spectrum the two models with the highest average projected NBP (CESM2 and NorESM2-LM) show a dominant contribution of sSM at the tropics mainly due to low values in the Amazon and high values in Indonesia (Figs. S7 and S9), as well as a positive contribution due to high long-term SM at the tropics and mid latitudes (see Fig. S12). Other noteworthy findings are: the strong contributions of sCO$_2$ and sSM at mid and high

latitudes to the higher-than-average land carbon sink projected by CNRM-ESM2-1 and IPSL-CM6A-LR; the less negative impact of hot temperature anomalies on tropical NBP for CMCC-ESM2 and MPI-ESM1-2-LR together with the relatively low long-term average tropical T of MPI-ESM1-2-LR, which result in positive contributions to average NBP; and the very low average NBP of CMCC-ESM2 at high latitudes resulting from steep drops during years with high T and low SM annual anomalies (sT and sSM contributions). These steep drops in annual NBP are associated with high fire emissions at the boreal

forests (Fig. S16), highlighting the importance of adequately representing this process in models given that it can explain much of the differences in average projected NBP. In this study, model differences in fire emissions are partly captured by differences in sT and sSM. Lastly, we note that CMCC-ESM2, GFDL-ESM4 and EC-Earth3-Veg show the largest uncertainties, particularly in the contributions of sT and sSM. This is related to high fire emissions from individual years which can be in or out of the bootstrap samples, as well as to partial shifts between bootstrap samples in the contributions of

sT and sSM due to collinearity.

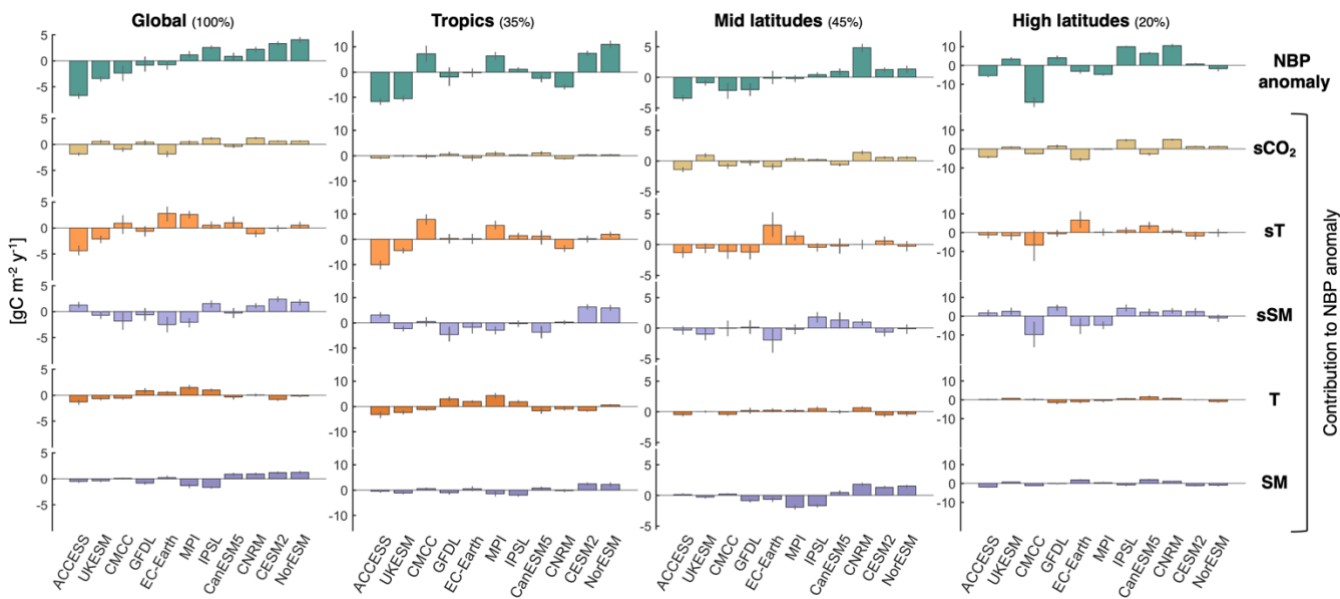

**Figure 7: Contributions of the drivers to explain the anomaly in average projected NBP from individual models relative to the ensemble mean.** Estimates are based on the multiple linear regression. Colored bars indicate the mean from the 200 bootstrap samples, while uncertainty bars span from the 5th to the 95th percentile. The spatial average is shown for global land, the tropics (22.5°S – 22.5°N),

mid latitudes (22.5°N – 47.5°N over North America, 22.5°N – 55°N over Europe and Asia, and > 22.5°S), and high latitudes (> 47.5°N over North America and > 55°N over Europe and Asia). The land percentage comprised in each region is noted next to the title: tropics represent 35.1% of the considered global land area, mid latitudes 44.8%, and high latitudes 20.1%.

In summary, the contributions of sT and sSM to explain the projected differences in NBP are the largest, whereas those of $sCO_2$, T and SM are generally smaller and of similar magnitude. The overall intermodel standard deviation of global land

projected NBP based on the regression estimates is 3.25 gC m$^{-2}$ y$^{-1}$ (equivalent to 37.8 PgC during the period 2015–2100), relative to which the intermodel standard deviations of the contributions from the drivers are 33.9% for $sCO_2$, 63.5% for sT, 52.7% for sSM, 27.1% for T and 31.2% for SM. These results highlight the importance of sSM and SM as drivers of

projected cumulative NBP, in addition to sT and T. Furthermore, the arguably high intermodel standard deviation values for many of the drivers suggest that while constraining any individual driver would help reduce the spread of the projected land

carbon sink, a large uncertainty would remain. For example, if we are to assume $sCO_2$ to be locally equal across all models, by summing the contributions from the other drivers we find that the intermodel standard deviation of NBP would still be 2.63 gC m$^{-2}$ y$^{-1}$, i.e., 81% of the original spread; assuming both T and SM to be locally equal across models drops the NBP spread to 80%; and assuming both sT and sSM to be locally equal across models drops it to 50%.

Figure 8 presents a compact overview of the factors explaining intermodel differences in cumulative projected NBP. We group sT with sSM to represent the sensitivity of NBP to interannual climate variability, and T with SM to represent general background climate conditions. This reduces any potential compensating effects in the contributions of sT, sSM, T and SM that could have resulted from the underlying collinearities between these drivers. Differences in the sensitivity of NBP to interannual climate variability play a key role, dominantly explaining the projected anomaly relative to the ensemble mean

for the two models with lowest land carbon sink, as well as for the two models with highest land carbon sink. We note here that ACCESS-ESM1-5 and UKESM1-0-LL (models with lowest sink) share a similar atmospheric model component (HadGEM family), while CESM2 and NorESM2-LM (models with highest sink) share the same land surface model. Furthermore, the intermodel variability in the contributions of the sensitivity of NBP to climate (sT+sSM) corresponds to 59% of the total NBP intermodel variability, whereas for the contributions of the $CO_2$ fertilization effect on GPP ($sCO_2$) it

corresponds to 33.9% and for the contributions of average climate conditions (T+SM) it is 28.3%. These insights are obtained based on the multiple linear regression, so it is worth noting once again that the regression estimates do not fully match the actual differences between models, with the clearest discrepancies in the underestimation of the sink for CanESM5 and NorESM2-LM, as well as in the overestimation UKESM1-0-LL.

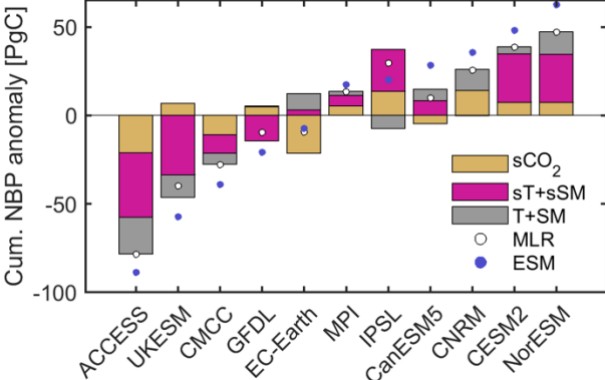

**Figure 8: Summary of contributions of the drivers to explain intermodel differences in projected global land cumulative NBP.** Bars indicate the grouped contributions of $sCO_2$, sT plus sSM, and T plus SM to the NBP anomaly estimate from the multiple linear regression. Dots indicate the total NBP anomaly based on the multiple linear regression (MLR) and from the actual model projections (ESM). All values correspond to averages from the 200 bootstrap samples. Global land estimates are obtained by multiplying the area-weighted averages times the land surface area excluding Antarctica (135.22E6 km$^2$). From left to right models are ordered according to actual

increasing projected land carbon sink.

The insights derived from Figure 8 are generally robust to different choices on how the sensitivities (sCO$_2$, sT and sSM) are defined. The estimated contribution of sCO$_2$ is larger when defining it as the sensitivity of NBP to rising CO$_2$, as opposed to the sensitivity of GPP to CO$_2$ (Fig. S17). In this case the intermodel variability in the contribution of sCO$_2$ increases from 33.9% to 43.5% of the total NBP intermodel variability, whereas that of sT+sSM decreases to 49.4% and that of T+SM decreases to 24.4%. This increase in the contributions of sCO$_2$ and decrease in those of sT+sSM is expected as model differences in the indirect effects of CO$_2$ on RA, RH and DIS are now included, in addition to a stronger collinearity of sCO$_2$ with sT and sSM, due to sCO$_2$ also capturing differences in RA, RH and DIS that are dependent on temperature and soil moisture. These indirect effects of CO$_2$ particularly contribute to explain the cumulative NBP differences of CanESM5, CNRM-ESM2-1 and IPSL-CM6A-LR relative to the ensemble mean. In addition, we replicate the analysis when computing sT and sSM from the 1pctCO2-rad simulations to remove potential alleviating effects of higher CO$_2$ given that these simulations account for the radiative effects of increasing CO$_2$ but keep CO$_2$ at the pre-industrial level from a biogeochemical perspective (Fig. S18). Results are still largely consistent with those of Fig. 8 even though this case is less meaningful as sT and sSM are computed under different CO$_2$ and climate conditions than those projected by scenario SSP126. Finally, results hardly change when assuming a latitudinal threshold of 30°, instead of 22.5°, beyond which annual mean warm season temperature and soil moisture are used to compute sT and sSM (Fig. S19).

## 6 Conclusions

In this study we focus on projections of the land carbon sink for a policy-relevant scenario with warming below 2°C by the end of the century (i.e., SSP126). Even under this scenario with a relative low concentration of greenhouse gases, there is an intermodel spread of approximately 150 PgC in cumulative NBP from 2015 to 2100 – equivalent to 15 years of current anthropogenic emissions – which translates into a 40% uncertainty in the carbon budget remaining to stabilize global temperature below the chosen threshold. We also show that even when two models project a similar global land carbon sink, there can be large and compensating regional differences. Here we identify regions in which models differ the most and assess which are the underlying model characteristics explaining these differences in the cumulative land carbon sink.

We accurately explain model differences in cumulative NBP as a function of their differences in the sensitivity of GPP to CO$_2$, in the sensitivity of NBP to interannual temperature and soil moisture variability, and in projected long-term temperature and soil moisture during the warm season. We detail differences in these five drivers across the model ensemble and discuss how they influence the land carbon sink projected by each model throughout the globe. In addition, we find that the detrended interannual variability of projected NBP is better explained by soil moisture than temperature in most models and across most regions. A notable exception is the core of the Amazon, where temperature is more important than soil moisture to explain the interannual variability of NBP in the models. Given the relevance of model differences in the sensitivity of NBP to temperature and soil moisture, it is increasingly important to further disentangle their sensitivities to

incoming radiation and vapor pressure deficit. This would bring us a step closer to identify the underlying mechanisms for the divergence across models from a process perspective.


Our quantification of the factors explaining intermodel differences in projected NBP highlights the dominant role of the response of the land carbon cycle to interannual temperature and soil moisture variability over that of the $CO_2$ fertilization effect and average climate conditions. This finding provides explicit evidence that improving the representation of the local-scale sensitivity of NBP to interannual climate variability has the potential to reduce uncertainty in long-term projections of

the global land carbon sink. A noteworthy aspect of this study is to explicitly consider the role of soil moisture when explaining model differences in projected cumulative NBP, as it provides valuable information in addition to temperature. We highlight substantial contributions mainly from model differences in the sensitivity of NBP to interannual soil moisture variability, but also from differences in long-term average soil moisture.

In the quest to better understand the future evolution of the land carbon cycle, our detailed insights about why each model projects either a relatively high or low cumulative land carbon sink is a valuable starting point to reducing uncertainty. For example, a high regional contribution of sSM and/or sT to a model's land sink anomaly indicates the need to evaluate and improve potentially related processes such as water stress on photosynthesis, the effect of temperature and moisture on soil carbon loss due to microbial activity, and the occurrence and magnitude of fire emissions. Furthermore, our findings

emphasize the need for spatially explicit observations of the sensitivity of the land carbon cycle to changes in temperature, soil moisture and $CO_2$ concentration, among other variables. Fortunately, this is becoming increasingly feasible thanks to progress in estimating carbon fluxes through in-situ observational networks, atmospheric inversions, and remote sensing products. The insights from this study together with those from novel observations are set to pave the way towards more confident projections of the evolution of the land carbon sink.

**Code and data availability**

The CMIP6 data used in this study are available at https://esgf-node.llnl.gov/search/cmip6/. Detailed inputs for the search query are as follows: Source ID: see Table S1; Experiment ID: ssp126, ssp585 and 1pctCO2-bgc; Frequency: mon; Variable: nbp, tas, mrsol, gpp, ra, rh, fFire. We also used the variable npp to derive ra = gpp – npp for the ACCESS-ESM1-5 model.

Scripts used for the analysis are available at: https://polybox.ethz.ch/index.php/s/iDyqY4lIrmIusUt.

**Author contributions**

RSP, LG, and SIS conceived the idea and designed the study. RSP processed the CMIP6 data, performed the analysis and wrote the paper with periodic suggestions from all authors. All authors discussed the results, read, and reviewed the paper.

**Acknowledgements**

We acknowledge partial support from the European Union's Horizon 2020 project "Climate-Carbon Interactions in the Current Century" (4C) under Grant Agreement No. 821003. VH acknowledges support from the Swiss National Science Foundation (grants no. P400P2_180784 and P4P4P2_194464). We acknowledge the World Climate Research Program's Working Group on Coupled Modelling, which is responsible for the Coupled Model Intercomparison Project (CMIP), and we thank the climate modeling groups for producing their model output and making it available. For CMIP, the US
Department of Energy's Program for Climate Model Diagnosis and Intercomparison provides coordinating support and led development of the software infrastructure in partnership with the Global Organization for Earth System Science Portals. We thank Urs Beyerle and Lukas Brunner for downloading and preprocessing the CMIP6 data.

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
