# Peer review of "Drivers of intermodel uncertainty in land carbon sink projections"

_Biogeosciences, 2022_

## Author Comment (AC1)

**Response to reviewers for BG-2022-92: "Controls of intermodel uncertainty in land carbon sink projections"**

*Referee #1:*

*Padrón et al report an analysis of drivers of the terrestrial carbon sink in the CMIP6 ensemble scenario SSP126 where warming is limited to 2 oC. This is a useful study of the latest CMIP model results in a policy relevant scenario, showing that terrestrial carbon sink projections by 2100 (cumulative NBP) in the ensemble vary from 56 to 207 Pg C, mean 144 and standard deviation 47 Pg C. Using linear regression Padrón et al partition this variability among sensitivity to CO2, temperature (T), soil moisture (SM), and differences in baseline temperature and soil moisture. Their methods show that the greatest proportion of this variance is explained by sensitivity to T and SM combined, with sensitivity to CO2 as the second most important driver of variability. Based on these results, they conclude that the gamma feedback (climate) is greater than the beta feedback (physiological) under this policy-relevant scenario and thus climate sensitivities require the greatest attention. They also show compensating drivers of cumulative NBP variability such that reduction of uncertainty in response to one driver would not greatly reduce overall NBP variability. Overall this is a well written and executed study. The analysis of the relatively low- warming SSP126 scenario is timely and to my knowledge has not been done before. I have several comments and criticisms that I hope will help to make the analysis and conclusions more robust and impactful. First, I think that for a number of reasons the method has low-biased the estimation of the impact of CO2 sensitivity on NBP variability. Second, I encourage a little more quantification and thought into exactly what is quantified and communicated*

We appreciate the overall positive evaluation of our manuscript.

To clarify, we do not conclude that the gamma feedback is greater than the beta feedback, but rather that model differences in the sensitivities of NBP to climate (related to gamma) contribute more to the land carbon sink intermodel spread than what model differences in the sensitivity of GPP to $CO_2$ (related to beta) do.

*First: underestimation of the impact of CO2 sensitivity on NBP variability. CO2 sensitivity is estimated as the sensitivity of GPP to CO2 in the 1 % per year increases in CO2 simulations (1pctCO2-bgc) in which CO2 ranges from 350 to 800 ppm. This method assumes 1) the CO2 sensitivity of NBP is the same as that for GPP, 2) that CO2 sensitivity is linear across the range 350 to 800 ppm, and 3) that there are no interactions between CO2 sensitivity and either T or SM. It is likely that all three of these assumptions will low- bias the estimate of the impact of model CO2 sensitivity on cross-model NBP variability.*

*While GPP sensitivity to CO2 is likely the main driver of NBP sensitivity to CO2, as asserted in the current ms, the assumption ignores potential changes in turnover rates that can also occur in response to CO2, which can be substantial. Using cross-model GPP sensitivity to CO2 will result in a lower correlation with NBP variability than using NBP sensitivity to CO2. Further, for T and SM sensitivity, NBP is used, biasing results in favor of T and SM sensitivity. Comparing the sensitivities of GPP to CO2 to NBP to T and SM is not a like-for-like comparison. Sensitivity of NBP to CO2 should be estimated and used in the regression analysis.*

We appreciate the insight, and now expanded the text to include this point. We also include in the supplement figures like Fig. 8 when using the sensitivity of NBP and NPP to $CO_2$ instead of the sensitivity of GPP for the analysis.

When using the sensitivity of NBP to $CO_2$, it is important to note that model differences can arise from (i) differences in the sensitivity of GPP to $CO_2$, (ii) differences in the sensitivities of RA, RH and DIS to $CO_2$ (i.e., the point of the reviewer), but also from (iii) differences across models in their sensitivities of RA, RH and DIS to temperature and soil moisture.

GPP generally increases with increasing $CO_2$ in the 1pctCO2-bgc simulations. This increase in GPP results in indirect effects of $CO_2$ on RA due to enhanced root respiration associated with greater belowground plant biomass, and on RH due to enhanced microbial decomposition of fresh carbon due to greater supply of foliar and root-derived labile soil carbon, and to increased microbial priming of old soil organic matter fueled by this increased supply of labile soil carbon (Gao et al., 2020). On the other hand, the magnitude of the changes in RA, RH and DIS in each model following the increase in GPP also strongly depend on the models' sensitivity of these fluxes to temperature and soil moisture (Todd-Brown et al., 2013; Varney et al., 2020).

The indirect effects of $CO_2$ on RA, RH and DIS are ignored when using $sCO_2$ as the sensitivity of GPP to $CO_2$, whereas the contribution of model differences in their sensitivity to climate are partially attributed to differences in $sCO_2$ when using the sensitivity of NBP of $CO_2$. To be comprehensive we now show the results of our analysis when using both the sensitivity of GPP and of NBP to $CO_2$.

Gao, Q., Wang, G., Xue, K., Yang, Y., Xie, J., Yu, H., Bai, S., Liu, F., He, Z., Ning, D., Hobbie, S. E., Reich, P. B. and Zhou, J.: Stimulation of soil respiration by elevated $CO_2$ is enhanced under nitrogen limitation in a decade-long grassland study, Proc. Natl. Acad. Sci., 117(52), 33317–33324, doi:10.1073/pnas.2002780117, 2020.

Todd-Brown, K. E. O., Randerson, J. T., Post, W. M., Hoffman, F. M., Tarnocai, C., Schuur, E. A. G. and Allison, S. D.: Causes of variation in soil carbon simulations from CMIP5 Earth system models and comparison with observations, Biogeosciences, 10(3), 1717–1736, doi:10.5194/bg-10-1717-2013, 2013.

Varney, R. M., Chadburn, S. E., Friedlingstein, P., Burke, E. J., Koven, C. D., Hugelius, G. and Cox, P. M.: A spatial emergent constraint on the sensitivity of soil carbon turnover to global warming, Nat. Commun., 11(1), 5544, doi:10.1038/s41467-020-19208-8, 2020

*The CO2 response over 350 to 800 ppm is likely not linear in these models, it almost certainly is not at the leaf scale which drives model CO2 responses. The SPP126 simulations max out at 446 ppm. There is likely saturation in the CO2 response for many models somewhere between 450 and 800 ppm. CO2 sensitivities should be estimated over the range of CO2 concentrations that preserve linearity over the range 350 – 446 (i.e. concentrations can be higher but responses must be linear over the range). A supplemental figure showing NBP against CO2 for the 1pctCO2-bgc simulations would be useful.*

We agree with this point. We expanded the text and now include the suggested figure in the supplement. We now use a range approximately between 375 and 500 ppm (the SSP126 range is 400 to 471 ppm) within which the response is relatively linear, while still counting

with a sample size of 30 annual values to estimate $sCO_2$ without being overly affected by internal climate variability. If we were to take fewer annual values from the single simulation for each model, particularly dry/wet and hot/cold anomalies during those years resulting from natural variability could lead to a biased estimate of $sCO_2$.

*There are interactions between CO2 and T and SM. Interactions with T are likely the most important for this discussion. At high T it is well known that CO2 can alleviate some of the reductions in photosynthesis due to interactive effects on photo- respiration. This could alleviate GPP reductions in high T years that I'm not sure would be removed by detrending NBP. I'm not sure there is an easy way to account for this, and that is OK. But some acknowledgment of this effect and some attempt to quantify it would help make results more robust.*

Interesting point. We now acknowledge it in the manuscript as another process that is only indirectly represented when trying to explain differences across models in their land C sink projections. Potential model differences in the alleviating effect of $CO_2$ at high temperatures are implicitly captured within differences in sT.

We additionally compute sT and sSM from the 1pctCO2-rad simulations to exclude the alleviating effect of $CO_2$ given that vegetation experiences constant $CO_2$ at pre-industrial level in these simulations. However, in this case we expect differences in sT and sSM compared to those derived from the SSP126 simulations, given that the background $CO_2$ concentration and climate conditions are different. We now also include in the supplement a Figure like Fig. 8 when using sT and sSM derived from the 1pctCO2-rad simulations.

*Second: encourage more thought into exactly what is quantified and communicated. I suggest quantifying statement in the abstract, cumulative NBP variability etc. Also, as well as putting these numbers in the context of current annual emissions, I think it might also be useful to present them as a proportion of the assumed emissions in the SSP126 scenario (if someone has calculated those).*

Thanks for the suggestions. We now include more quantitative statements in the abstract. We also indicate that the intermodel spread of ~150 PgC corresponds to approximately 40% of the remaining carbon budget to limit global warming below 2°C (with a 50% likelihood) according to Table SPM.2 of IPCC (2021).

IPCC, 2021: Summary for Policymakers. In: *Climate Change 2021: The Physical Science Basis. Contribution of Working Group I to the Sixth Assessment Report of the Intergovernmental Panel on Climate Change* [Masson-Delmotte, V., P. Zhai, A. Pirani, S.L. Connors, C. Péan, S. Berger, N. Caud, Y. Chen, L. Goldfarb, M.I. Gomis, M. Huang, K. Leitzell, E. Lonnoy, J.B.R. Matthews, T.K. Maycock, T. Waterfield, O. Yelekçi, R. Yu, and B. Zhou (eds.)]. Cambridge University Press, Cambridge, United Kingdom and New York, NY, USA, pp. 3−32, doi:10.1017/9781009157896.001.

*Why is the proportion of variance in NBP variability to CO2 sensitivity not quantified on ln 354?*

We now rephrased the sentence to include this.

*I encourage the authors to think about what is best to present given this is a study of the global carbon cycle. Most figures are presented in the units per meter squared. When aggregating to broad zonal regions I suggest it is more informative to present results as the absolute sum across the whole area – this would make it easier to relate the regions and sensitivities directly to the global aggregate numbers. Finally, have differences in grid-square area been taken into account when presenting the global aggregate drivers of NBP variability?*

Thanks for the suggestions. We modified Fig. 8 to express the units in carbon mass for the whole land area. For Figs. 6 and 7 expressing the data in $gC\ m^2\ y^{-1}$ helps the visualization, and our goal of comparing models and drivers rather than comparing across regions. Nonetheless, we now make it more visible in the figures what percentage of the land area corresponds to each region instead of only mentioning this in the caption. Finally, we do consider differences in grid cell area according to latitude.

*Technical Comments:*

*Title: suggest switching "controls" for "drivers" as control suggests some degree of intention.*

We incorporated this suggestion.

*While I see some of the benefits of the narrative style with the methods spread throughout the results (e.g. lns 157-164, 179-189, 263-283, etc), I think it is more practical to have the methods all in one place where they can be found easily and assessed side by side.*

Thanks for the suggestion. We consider this to be a matter of style and given that typically papers in *Biogeosciences* do not have a specific "Methods section", we would prefer to keep the original narrative style.

*Fig 6: can probably go into the supplement.*

We appreciate the suggestion, but nonetheless consider it is useful to keep Fig. 6 in the manuscript to convey the message that the regression estimates can represent well intermodel differences in regional projected NBP.

*Fig 7: A little hard to read, I think the sensitivities could be presented more clearly if they were presented as in Fig 8. I recognise that would necessitate removal of uncertainties from the figure and I appreciate the effort made to quantify uncertainty but is clarity of communication is the trade off.*

We now also present the results from Fig. 7 as stacked bars in the supplement.

*Fig 8 as is could go to the supplement. Fig 8: Suggest adding a dot for the actual cumulative NBP. Also I really think this would be better off expressed in global sums rather than per meter squared. The white dot could be a little larger.*

We included the suggestions.

*Ln 30: There are several commentaries explaining why Wang et al 2020 is not a reliable analysis.*

We decided to remove this reference from the text.

*Ln 37: Are there other disturbances that release C directly back to the atmosphere?*

Not directly, but other disturbances can include changes in land cover like deforestation.

*Ln 61: Note the editor's note for Keenan 2021*

Thanks for pointing this out. We removed this reference, as the article has been retracted.

*Ln 69: can delete "consider it important to instead"*

We rephrased the sentence.

---

## Author Comment (AC2)

**Response to reviewers for BG-2022-92: "Controls of intermodel uncertainty in land carbon sink projections"**

*Referee #2:*

*This manuscript takes 11 ESMs from the CMIP6 archive and attempts to unravel the root causes of the uncertainty in the NBP simulated by each model over the (roughly) 2 deg warming scenario. The models' sensitivity to CO2 (through the 1 percent runs) and to temperature and soil moisture are investigated for both short and long timescales. I found the paper figures to be generally well designed (though see my comment about Fig 3) but the text could be confusing at times. There is a lot of rather convoluted steps/arguments in producing the T/SM/sT/sSM metrics and it sometimes was hard to understand exactly what they were telling me about the models. I think the paper is publishable, but needs revisions for clarity. An obvious target for clarity/context would be to discuss the results of this work in the context of previous efforts as discussed in the introduction (principally Arora et al. 2020). I found myself comparing this work to that paper and not understanding why they differed strongly in some cases.*

We appreciate the overall positive evaluation of our manuscript. We rephrased some sentences in the text to clarify what do sT and sSM represent and to explain their effect as drivers of differences in cumulative NBP.

Our analysis is complementary to that of Arora et al. (2020) by focusing on the relevance of soil moisture and temperature for land carbon sink projections. Moreover, we provide more regional insights and address projections under a low emission scenario. Thus, most of our results are not directly comparable to those of Arora et al. (2020). Nevertheless, when possible, we don't find strong differences with the insights from Arora et al. (2020) on why each model projects either a relatively low or high land C sink. For example, the contribution of $sCO_2$ ($\Delta GPP/c'$) is similar in both studies for explaining differences across models. We believe there was a slight misunderstanding of two statements in our manuscript, as discussed in our reply to other comments below.

*Main comments:*

*Fig 3: I found this to be a strange figure. So if a model has a positive correlation it counts towards the blue end of the colour scheme, whereby if it is negative it counts towards the red. However this seems to have no consideration of how positive or negative a model was. I think it would treat a model that is 0.9 the same as one that is 0.0009, which seems to be a bit too ambiguous. Also what if all 11 models are +0.0001 vs. all models are >0.9, as is they would appear the same in this figure but arguably the situation where all 11 are >0.9 is more interesting than the situation where the models are rather ambiguous (close to 0). I would suggest reconsidering this figure.*

We modified the figure and now show the ensemble mean correlation divided by the standard deviation to better illustrate the confidence in the sign of the relation and the intermodel discrepancy. The full picture of the actual correlations for each model are provided in Figs. S5 and S6.

*Deserts - how did you mask the deserts? I think the grid cell sizes of the ESM precludes removing many deserts, e.g. the Atacama. Instead it seems like only a few were removed*

*(Sahara, around Middle East, and Gobi) but I am not sure why those made the cut but not, for example the deserts of western Australia or the US SW. What impact does it have keeping them in? Greenland makes sense since there is no vegetation at all but some of the world's dry regions have been fingered as influential in the global C cycle (e.g. Ahlström et al. 2015), so exactly where masking applies could have impact I would assume.*

Thanks for pointing this out. Masked grid cells correspond to those with observed annual mean precipitation below 100 mm based on the GSWP3 dataset for the period 1985–2014. The resolution of the grid indeed plays a role in which grid cells are masked. The transition between semi-arid and desert regions likely depends on the resolution of each model. Precisely because of this, we regridded all models to the same grid and omitted the same grid cells to make them more comparable.

We now include in the supplement a Figure like Fig. 8 when not masking desert grid cells.

*Fire - Fire is mentioned on line 335 but ignored otherwise, why? I see you mention which models do fire in Table S1.*

We also refer to model differences in disturbance fluxes in the manuscript and in Fig. S2, but now clarify that this is strongly related to fire. We now also clarify that sT and sSM may indirectly account for model differences in fire emissions.

*Smaller comments:*

*Line 44: 'with drought-related observed decreasing trends in leaf area' consider rewording, confusing.*

We rephrased the sentence for clarity.

*Line 103: CanESM has an implict N cycle (empirical downregulation scheme see Arora and Scinocca 2016)*

Thanks for pointing this out. Our original statement is based on Table 2 from Arora et al. (2020), where it is mentioned that CanESM5 has no representation of the N cycle.

*L 160: This explanation is confusing. Perhaps spell it out in a bit more detail.*

We expanded the text for clarity.

*L 182: Why 22.5 degrees and not 25 or 30 or some other number? It just seems awfully precise for a seemingly arbitrary limit.*

Tropical regions are defined as those with latitudes lower than 23.43 degrees. Given that are employed grid has a resolution of 2.5 degrees, we choose the threshold of 22.5 degrees as the nearest one to the definition of tropical regions. We now also include in the supplement an additional figure when using a threshold of 30 degrees to distinguish which months are considered as the warm season.

*L 235: remind reader that both use CLM?*

We now also mention here that both models have CLM5 as their land surface model.

*L 290: 'underestimation of the land carbon sink modelled by NorESM2-LM and CanESM5,' where is this shown? I can't seem to see any figure where CanESM5 sticks out with an underestimation of the land C sink but it is mentioned here and line 357, indeed in Figure 1 it seems to have one of the highest cumulative NBP. What am I missing?*

This is referring to the comparison between the regression estimate and the projected land C sink from the ESM as shown in Fig. 6. In this paragraph it is not about comparing the land C sink across models. We expanded the text to clarify this point.

*CanESM5: Other papers (Arora et al. 2020) have suggested that CanESM5 has the largest land C uptake (at least for the 4X CO2 simulations) so it is surprising that it is suggested to be underestimated for the land C sink. Can you clarify how the same model appears to be on the low/high end depending on the analysis? I realize these are different scenarios but I would have assumed high CO2 sensitivity would follow in both (but be exaggerated in the 4XCO2 run), but I don't see high CO2 sens in Fig 7. I assume I missed something here as you mention the Arora et al. paper in the intro but don't return to place your results in context of those other works.*

There is a slight misunderstanding. We do not suggest that CanESM5 underestimates the land C sink compared to other models. As mentioned in the reply above, in that part of the manuscript we just note that the regression estimate of the land C sink for CanESM5 is lower (underestimated) than the actual land C sink projected by CanESM5.

It is indeed the case that CanESM5 shows a very high land C sink compared to other models, particularly while $CO_2$ concentrations and temperatures are rising until approximately the year 2070 as shown in Fig. 1. Northern mid and high latitudes strongly contribute to the high land C sink in CanESM5 as shown in Fig. 2. Our findings suggest that the predominant reason for this high land C sink in CanESM5 is its sensitivity of NBP to T (sT in Fig. 7), rather than its $sCO_2$. It is clear that CanESM5 has a more positive correlation of NBP and T over the Northern hemisphere compared to other models (Fig. S5), whereas its $sCO_2$ is not at the high end of the ensemble (Fig. S4). These findings are consistent with the results from Fig. 8 in Arora et al. (2020) where $CUE_\Delta$ (somewhat related to sT) contributes more than $\Delta GPP/c'$ (related to $sCO_2$) to the high land C sink projected by CanESM5.

*L 352: And assumedly many of them use Nemo for their ocean so model commonalities are not just atm/land. Never mind all who use Farquar photosynthesis etc*

This is true. However, the purpose of our statement is just to note that the two models with the lowest projected land C sink share the same atmospheric model, and the two with the highest projected sink share the same land model. We rephrased the sentence to clarify this.

*L 378: 'Outperfom' seems out of place, consider swapping it out with something like 'be more important than'*

Thanks for the suggestion. We rephrased the sentence.

*Lit cited:*

Ahlström, A., Raupach, M. R., Schurgers, G., Smith, B., Arneth, A., Jung, M., Reichstein, M., Canadell, J. G., Friedlingstein, P., Jain, A. K., Kato, E., Poulter, B., Sitch, S., Stocker, B. D., Viovy, N., Wang, Y. P., Wiltshire, A., Zaehle, S., and Zeng, N.: The dominant role of semi-arid ecosystems in the trend and variability of the land CO2 sink, Science, 348, 895–899, 2015.

Arora, V. K., Katavouta, A., Williams, R. G., Jones, C. D., Brovkin, V., Friedlingstein, P., Schwinger, J., Bopp, L., Boucher, O., Cadule, P., Chamberlain, M. A., Christian, J. R., Delire, C., Fisher, R. A., Hajima, T., Ilyina, T., Joetzjer, E., Kawamiya, M., Koven, C. D., Krasting, J. P., Law, R. M., Lawrence, D. M., Lenton, A., Lindsay, K., Pongratz, J., Raddatz, T., Séférian, R., Tachiiri, K., Tjiputra, J. F., Wiltshire, A., Wu, T., and Ziehn, T.: Carbon–concentration and carbon–climate feedbacks in CMIP6 models and their comparison to CMIP5 models, Biogeosciences, 17, 4173–4222, 2020.

Arora, V. K. and Scinocca, J. F.: On constraining the strength of the terrestrial CO2 fertilization effect in an Earth system model, https://doi.org/10.5194/gmd-2015-252, 2016.

---

## Author Response (AR2)

**Response to reviewers for BG-2022-92: "Drivers of intermodel uncertainty in land carbon sink projections"**

List of most relevant changes made in the manuscript:

- We follow the reviewer's suggestion and now use the sensitivity of NBP to $CO_2$ ($sCO_2$) instead of the sensitivity of GPP to $CO_2$ as one of the drivers of intermodel uncertainty in land carbon sink projections. We therefore updated Figures 5 to 8, as well as parts of the text.

*Referee #1:*

*Padrón et al argue that in a multiple regression of the drivers of cumulative NBP, the explanatory variable related to CO2 should be based on the sensitivity of GPP to CO2 and not NBP as follows (Ln 162 – 168): "These simulations limit confounding effects from changes in temperature and soil moisture as they only account for the biogeochemical effects of rising CO2. However, when computing the change in NBP in these simulations, it is important to note that model differences can also arise from differences in RA, RH and DIS that are highly dependent on how these fluxes are influenced by temperature and soil moisture in each model. Therefore, we decide to use the sensitivity of GPP (instead of NBP) to CO2 as a driver of intermodel uncertainty in land carbon sink projections to better disentangle the influence of CO2 from that of temperature and soil moisture, even though the indirect effects of CO2 on RA, RH and DIS are ignored in this case."*

*I don't agree with this line of reasoning. In the 1pctCO2-bgc simulations there is no radiative coupling to increasing CO2 so there is no radiatively-driven trend in climate in these simulations. Thus the trend in NBP with CO2 should not be affected by trends in T and SM as there are no trends in T and SM unless affected through the physiological action of CO2 on stomatal conductance. I assume the authors are arguing that differences in model baseline T and SM may influence NBP in the 1pctCO2-bgc, which I guess they do, but also assume their influence on the response of NBP to CO2 is small. And baseline differences in model T and SM are accounted for in the cumulative NBP multiple regression already.*

We appreciate the comment. Our main point is rather that there are important model differences in the sensitivity (i.e. the slope of the regression) of RA, RH and DIS to interannual temperature and soil moisture variability, which can explain differences in projected cumulative NBP from the 1pctCO2-bgc simulations due to the asymmetric nature of the response of NBP to cold/wet and dry/hot years. Therefore, we expect some additional collinearity between sT and sSM with $sCO_2$ when computing $sCO_2$ as the sensitivity of NBP to $CO_2$. This was our primary reason to use $sCO_2$ as the sensitivity of GPP to $CO_2$ in the main text of the manuscript. Nevertheless, we agree with the comments below and now use throughout the main text the sensitivity of $CO_2$ to NBP as a driver of the intermodel uncertainty in land carbon sink projections.

*My original point stands that the various drivers of inter-model spread in cumulative NBP are not compared on an equal footing. While sT and sSM are NBP sensitivities, sCO2 is a GPP sensitivity. This is illustrated by the analysis in the supplement of sCO2 is calculated using NBP instead of GPP (compare Figure S17 a and b respectively). In almost all cases the multiple regression prediction of cumulative NBP is closer to the ESM cumulative NBP when sCO2 is calculated using NBP rather than GPP (Figure S17). In some cases the change is small (but never worse), while in some cases the improvement is substantial – e.g. the white and blue points in S17a are closer than in S17b for ACCESS, IPSL, CanESM, CNRM.*

*This is an important point because for almost all models, using NBP to calculate sCO2 also increases the proportion of cumulative NBP that is attributable to sCO2, i.e. their CO2 sensitivity (in addition to improving the multiple-regression model fit). It's not clear by how much from the presentation of*

*the results but it seems like differences in model CO2 sensitivities are of similar magnitude as T and SM sensitivities at explaining inter-model spread in cumulative NBP.*

*The CO2 sensitivity needs to be calculated with NBP as the response variable, not GPP. This will require a major revision of some of the text and figures.*

We agree with this insight. We now replace Fig. 8 with Fig. S17 and modify the text accordingly.

*Ln 16-23: "Results indicate a primary role of the response of NBP to interannual temperature and soil moisture variability, followed by the sensitivity of photosynthesis to CO2, and lastly by the average climate conditions, which also show sizeable contributions. We find that the sensitivities of NBP to temperature and soil moisture, particularly in the tropics, dominantly explain the deviations from the ensemble mean of the two models with the lowest carbon sink (ACCESS-ESM1-5 and UKESM1-0-LL) and of the two models with the highest sink (CESM2 and NorESM2-LM). Overall, this study provides insights on why each Earth system model projects either a low or high land carbon sink globally and across regions relative to the ensemble mean, which can focalize efforts to identify the representation of processes leading to intermodel uncertainty."*

*Three of these highest and lowest models have a significant shortfall in prediction, possibly due to interactions or drivers missing from the regression.*

We agree with this point. Another possibility is that this underestimation of the magnitude of the anomalies occurs due to a non-linear response to the sensitivities which are not captured by the multiple linear regression. This is discussed in the main text. We consider it too detailed to mention it in the abstract. We also note that this shortfall in prediction does not contradict our conclusions.

*These results and conclusions presented in the abstract need to be a lot more quantitative. E.g. Why not quantify the contribution of each driver to inter-model spread rather than use language like "which also show sizeable contributions."*

We modified the abstract accordingly.

*Before I can recommend this for publication, sCO2 should be calculated with NBP not GPP as the response variable and used in the multiple regression and other areas of the manuscript.*

We now follow this suggestion.

*Editor:*

*The authors are arguing to use the sensitivity of GPP to CO2 and not NBP. My feeling is that modelled GPP itself would be subjected to marked differences in the temperature sensitivity (Rogers et al. 2017 New Phytologist) and soil moisture sensitivities (De Kauwe et al. 2017 Global Change Biology) that act on GPP. As a result, I'm not as convinced that this limits confounding effects. I think the authors need to address R2's comment as it is important point.*

We appreciate the comment and now also mention it in the text.